∂ | Open Peer Review | Bacteriology | Research Article

# Amoeba plate test with *Acanthamoeba castellanii* as an innovative tool for *Nocardia* recovery from sputum samples: a proof-of-concept study

Coralie Espinasse,[1,2] Ghislaine Descours,[1,2,3] Marine Ibranosyan,[1,2] Laetitia Beraud,[1] Christophe Ginevra,[1,2,3] Joelle Chastang,[1] Oana Dumitrescu,[4,5] Frédéric Laurent,[4,6] Verónica Rodriguez Nava,[5,7] Sophie Jarraud,[1,2,3] Camille Allam[1,2]

**ABSTRACT**  *Nocardia* recovery from pulmonary samples is challenging due to the lack of a specific medium and the abundance of overgrown respiratory flora. This study aimed to compare the amoeba plate test (APT), an amoebic coculture with *Acanthamoeba castellanii*, with the axenic culture to recover *Nocardia* from pulmonary samples. *Nocardia* serial dilutions ($n = 15$ strains from seven species, concentrations ranging: $10^7$–$10^2$ CFU/mL) in water and spiked overgrown sputa ($n = 8$) were simultaneously plated on agar with amoebic monolayer (APT) and without (control). Culture positivity rates, minimal growth concentrations, growth times, and abundance of flora overgrowth were compared for each condition. In water, *Nocardia* culture positivity rates were not significantly different between APT (86%, 30/35) and control (94%, 33/35; $P = 0.246$). In sputa, APT resulted in greater *Nocardia* growth (63%, 22/35 vs 37%, 13/35; $P = 0.008$). In addition, the elimination of interfering flora was more frequent using APT (34%, 12/35 vs 11%, 4/35; $P = 0.01$), and the overall abundance of flora was lower (median [interquartile range, IQR]: 1 [0–2] vs 3 [1-3]; $P < 0.0001$). The most grown species using APT were *N. mexicana* and *N. otitidiscaviarum,* whereas *N. abscessus* and *N. nova* were the most fastidious to grow under both conditions. This study reports *Nocardia*'s ability to grow in amoebic coculture, with some growth kinetic differences depending on the species, presumably implying their intra-amoebic multiplication. Furthermore, in APT, *Nocardia* recovery from overgrown sputa was greater, and flora decontamination was more effective. The present findings are strong arguments to implement APT as a complementary technique for *Nocardia* isolation from heavily contaminated samples.

**IMPORTANCE**  The culture-proven diagnosis of Nocardiosis is challenging due to the difficulties in recovering *Nocardia* colonies from respiratory samples containing a complex polymicrobial flora. However, the isolation of *Nocardia* strains remains necessary to perform antibiotic susceptibility testing and adapt the antibiotic regimen of the patients. This study provides an innovative culture method based on a solid medium amoebic coculture, the amoeba plate test (APT), to cultivate *Nocardia* strains from clinical sputa samples. *Nocardia* grew across the APT amoebic monolayer. Moreover, the APT, which is already used in a reference center for the recovery of *Legionella* strains, exhibited good performances for *Nocardia* recovery and decontamination of interfering flora, thereby representing a promising second line tool to improve *Nocardia* culture.

**KEYWORDS**  Nocardiosis, *Nocardia*, amoebae, amoeba plate test, decontamination, culture diagnosis

M embers of the genus *Nocardia* are responsible for pulmonary Nocardiosis, an underreported disease (1–3) found in immunocompromised and

**Peer Reviewer** Lucio Vera-Cabrera, Universidad Autonoma de Nuevo Leon, Monterrey, Mexico

Address correspondence to Sophie Jarraud, sophie.jarraud@chu-lyon.fr.

The authors declare no conflict of interest.

immunocompetent subjects (4, 5). There is a high mortality rate among patients with disseminated Nocardiosis, cavitary pulmonary infection, or pleural effusion (6). The lack of *Nocardia* screening methods (1, 4) and the varied clinical presentation (2) are the main causes of Nocardiosis underdiagnosis. Although molecular biology techniques have been developed, the definitive diagnosis and the antibiotic susceptibility testing (AST) rely on *Nocardia* positive culture from pulmonary samples (sputa, trachea-bronchial aspirates, or bronchoalveolar lavages) (7, 8). Samples are usually cultured on enriched media (Middlebrook 7H10 or buffered charcoal yeast extract, BCYE) (2, 8), and no selective medium for *Nocardia* is commercially available. Depending on the bacterial load and the *Nocardia* species, colonies grow from 2 to 3 days to 10 days or more (9), and their isolation from pulmonary samples is a low-effective procedure (2, 10); fast growing commensal or pathogenic microorganisms may overgrow on the medium instead of *Nocardia*.

A few studies reported on the intracellular multiplication of *Nocardia* in animal macrophages and brain cells (11–13) or in murine cell lineages (14). *Nocardia* are ubiquitous microorganisms found in natural environments, including soils and waters (2, 4), where other unicellular organisms, such as free-living amoebae (FLA), are also found. However, their ability to grow within amoebae remains currently unknown. On the contrary, it is well-established that *Legionella* have an intra-amoebic life cycle in the environment (15, 16); this intra-amoebic ability to grow has been used in laboratory routine practice for *Legionella* isolation from pulmonary samples with bacterial overgrowth using an amoebic and axenic coculture (17). The amoeba plate test (APT) adapted from liquid amoebic coculture relies on the *Legionella* growth across an amoebic monolayer plated on a solid medium and on the elimination of interfering bacteria by amoebae. Combined to axenic culture, APT adds value to recover *Legionella* strains (18).

In this context, the present study aimed to evaluate both *Nocardia's* ability to grow in the presence of amoebae using the APT and the APT performance to recover *Nocardia* strains from pulmonary samples containing a bacterial overgrowth.

## MATERIALS AND METHODS

### *Nocardia* strains

A total of 15 strains from the following seven species or complexes of *Nocardia* from the collections of the *Centre de Ressources Biologiques Environnement Microbiologie Lyon* (EML, strain collection belonging to *Observatoire Français des Nocardioses*) or of *Deutsche Sammlung von Mikroorganismen* (DSM) were used: *N. wallacei* (EML 1472, 1473, and 1474), *N. mexicana* (EML 1475 and 1476), *N. otitidiscaviarum* (EML 1477, DSM 43242$^T$), *N. farcinica* (EML 1478 and 1479), *N. cyriacigeorgica* (EML 1480), *N. abscessus complex* (DSM 44432$^T$, EML 1481 and 1482), and *N. nova complex* (DSM 44481$^T$, EML 1483). All strains were identified by sequencing *rrs* and *hsp65* genes before the assays (19). All strains were stored in cryotube at −80°C and grown on 5% blood sheep agars (COS, BioMérieux, Marcy-L'étoile, France) at 30°C for 72 h before the assays.

### *Nocardia* dilutions—sterile water

Suspensions of *Nocardia* were prepared by adjusting 600 nm optical density (OD) at 0.1 unit in sterile water, corresponding to a concentration of $10^8$ CFU/mL. From this initial suspension, 10-fold serial dilutions were carried out in sterile water (Sigma-Aldrich, Saint Louis, MO, USA) to obtain six dilution points ranging from $10^7$ to $10^2$ CFU/mL. Suspensions were homogenized prior to OD measurement and between the dilution points by vortexing for 30 s and pipetting. Since *Nocardia* usually grow as entangled filaments, particular attention was paid to obtain a homogeneous, precipitate-free suspension prior to OD measurements.

Depending on the strains, one to three experimental replicates were performed for APT and control (Fig. S1; Table S1).

## *Nocardia* overload—pulmonary samples

A total of 50 µL of each *Nocardia* dilution previously obtained in sterile water was added to 50 µL of one sputum sample to obtain six dilution points ranging from $10^7$ to $10^2$ CFU/mL (Fig. S1). Each sputum sample was used for the extemporaneous preparation of two to nine different *Nocardia* strain suspensions (Table S2). Sputa were remnants from the microbiological laboratory routine (Institut des Agents Infectieux, Hospices Civils de Lyon, France) after axenic culture for diagnosis purposes and selected according to the following criteria: recent samples (<7 days) with significant growth (≥$10^7$ CFU/mL) of one or several bacterial species. Those with growth of numerous yeasts or molds were excluded.

Samples were previously fluidified using a volume-to-volume digest-EUR (Eurobio, Les Ulis, France) solution and stored at 4°C before the assay. The bacterial species that grew from the sputa cultures for diagnosis purposes were previously identified by the microbiology laboratory using mass spectrometry (matrix-assisted laser desorption/ionization time-of-flight, Vitek MS, BioMérieux, Table S2) from Columbia CNA (colimycine, nalidixic acid) supplemented with 5% sheep blood (BioMérieux) and chocolate agars (PolyViteX, BioMérieux) incubated for 48 h in 5% CO2 and Mac Conkey (BioMérieux) agar incubated for 48 h in an aerobic atmosphere. The associated commensal oropharyngeal flora (OPF), including oropharyngeal *Streptococci*, coagulase negative *Staphylococci*, *Neisseria* spp., and commensal Gram-negative bacilli, was presumptively identified based on macroscopic culture characteristics only. The bacterial flora (BF) was all the cultivable bacteria, OPF and/or pathogenic, that were found in sputa.

## Amoeba plate test

A fresh culture of *Acanthamoeba castellanii* (ATCC 30234) was inoculated on BCYE supplemented with 4 mg/L cefamandole, 80,000 UI/L polymyxin B, and 80 mg/L anisomycin (BMPA; Oxoid, Dardilly, France) according to a described protocol for *Legionella* amoebic coculture (18). The APT was prepared twice a week. Amoebae were maintained in peptone yeast extract–glucose medium at 30°C for 3 days in a 75 cm$^2$ flask. Confluent amoebae were recovered by manual shaking in 10 mL of Page's amoebic saline (PAS). Amoebae were counted from the PAS suspension. The appropriate volume of suspension for a final quantity of $3.75 \times 10^6$ amoebae/APT was centrifuged for 5 min at 500×*g*. The supernatant was then removed, and the pellet was resuspended in PAS to obtain a final concentration of $2.5 \times 10^6$ amoebae/mL. Next, 1.5 mL of this suspension was plated on each BMPA with a rake. The amoebae for APT were dried under a laminar flow hood for 1 h before inoculation. Then, 3 µL spots of previous *Nocardia* dilutions in sterile water or in sputa were inoculated on each APT and on BMPA used as a control medium (*n* = five replicates for each dilution). The amoebae monolayer viability was verified using a suspension (OD = 1, 600 nm) of a previously described *L. pneumophila* reference strain (CIP 107629T) with *dotA* deleted, deficient for intra-amoeba replication (18, 20). The BF viability in the APT and control was verified by spotting 3 µL of each sputum (*n* = five replicates) without *Nocardia* overload. The APT and control plates were incubated at 30°C in 2.5% $CO_2$ and analyzed every 2 days for 10 days. *Nocardia* colonies on the APT and control were visually detected by their macroscopic aspect (e.g., brain-like white to orange colonies with a smooth to chalky aspect). The day of culture positivity and the minimal growth concentration (MGC) were assessed for each replicate on the APT and control media. The MGC was defined as the lowest suspension concentration that led to a positive result in culture. The BF growth was visually detected based on macroscopic culture characteristics, and its abundance on the spots was estimated using an ordinal scale: 0 = absent; 1 = low (≤5 colonies), 2 = medium (6–50 colonies), and 3 = high (>50 colonies).

## Statistics

The percentages of *Nocardia* and/or BF-positive cultures for the APT and control were compared using paired-samples McNemar tests. MGC, expressed as medians for each species replicate (Tables 1 and 2), and median days of culture positivity were compared using Mann–Whitney tests. Median BF abundances were compared using Wilcoxon rank-sum tests. The comparison of medians was only performed for assays with a positive culture through Mann–Whitney and Wilcoxon rank-sum tests using R 4.3.3 (Vienna, Austria). McNemar tests were performed for all the assays using R 4.3.3. For all statistical tests, the *P*-value threshold for significance was set at 0.05. A graphical representation of *Nocardia* and BF growth percentages, as well as BF abundances were obtained using GraphPad Prism software (San Diego, CA, USA).

## RESULTS

### *Nocardia* grow across the amoebic monolayer

Serial dilutions of *Nocardia* suspensions (Table 1) were spotted on APT and control (example of *Nocardia* growth in APT and control in Fig. 1). There was no significant difference regarding the rates of *Nocardia*-positive cultures between the APT (86%, 30/35) and control (94%, 33/35, *P* = 0.248; Table 1). The MGC were not different between the APT and control, ranging from $10^3$ to $10^5$ CFU/mL depending on the species (Table 1). The *N. abscessus* complex was the most fastidious to cultivate compared to the six other species either using the APT (MGC: $10^4$ CFU/mL vs $10^3$ CFU/mL, *P* = 0.021) and control (MGC: $10^5$ CFU/mL vs $10^4$ CFU/mL, *P* = 0.007). *Nocardia* growth using APT was obtained within a median (IQR) of 2 (2–4) days without significant difference compared to control. *N. mexicana* and *N. otitidiscaviarum* were the most rapid to grow (2 [2–2] and 2 [1–2] days, respectively), while *N. abscessus* and *N. nova* were the less rapid (4 [3–6] and 5 [4–6] days, respectively; Table 1). Details of individual strain growth are presented in Table S1.

### APT shows a high sensitivity for *Nocardia* recovery from polymicrobial clinical pulmonary samples

To mimic clinical conditions, the same *Nocardia* serial dilutions were added to eight sputa with a high level of BF (e.g., $\geq 10^7$ CFU/mL of pathogenic bacteria and/or of bacteria

**TABLE 1** APT and control culture positivity rates, MGC, and growth times for *Nocardia* in sterile water[a]

| *Nocardia* species | Culture positivity rate (n/n) | | | MGC[b] (median [IQR], CFU/mL) | | | Growth time (median [IQR], days) | | |
|---|---|---|---|---|---|---|---|---|---|
| | APT | Control | *P* value | APT | Control | *P* value | APT | Control | *P* value |
| *N. wallacei* (n = 7) | 7/7 | 7/7 | | $10^3$ [$10^3$–$10^4$] | $10^3$ [$10^3$–$10^4$] | | 2 [2–3] | 2 [2–2] | 0.178 |
| *N. mexicana* (n = 5) | 5/5 | 5/5 | | $10^3$ [$10^3$–$10^3$] | $10^3$ [$10^3$–$10^3$] | | 2 [2–2] | 2 [1–2] | 0.600 |
| *N. otitidiscaviarum* (n = 4) | 4/4 | 4/4 | | $10^3$ [$10^3$–$10^3$] | $10^3$ [$10^3$–$10^3$] | | 2 [1–2] | 1 [1–1] | 0.608 |
| *N. farcinica* (n = 5) | 5/5 | 5/5 | | $10^3$ [$10^2$–$10^3$] | $10^4$ [$10^3$–$10^4$] | 0.519 | 3 [3–4] | 2 [2–3] | 0.273 |
| *N. cyriacigeorgica* (n = 3) | 3/3 | 3/3 | | $10^3$ [$10^3$–$10^3$] | $10^3$ [$10^3$–$10^4$] | 0.307 | 2 [2–5] | 2 [2–3] | 1 |
| *N. abscessus* complex (n = 7) | 3/7 | 5/7 | 0.480 | $10^4$ [$10^4$–$10^5$] | $10^5$ [$10^5$–$10^5$] | 0.637 | 4 [3–6] | 4 [4–5] | 0.814 |
| *N. nova* complex (n = 4) | 3/4 | 4/4 | 1 | $10^4$ [$10^4$–$10^6$] | $10^4$ [$10^3$–$10^4$] | 0.643 | 5 [4–6] | 4 [3–4] | 0.658 |
| Total (n = 35) | 30/35 | 33/35 | 0.248 | $10^3$ [$10^3$–$10^4$] | $10^4$ [$10^3$–$10^4$] | 0.146 | 2 [2–4] | 2 [2–3] | 0.263 |

[a]CFU: colony forming unit; APT: amoebae plate test; MGC: minimal growth concentration; IQR: interquartile range.
[b]MGC is the median of the replicate lowest suspension concentration that gave a positive result in culture.

**TABLE 2** APT and control culture positivity rates, MGC, and growth times for *Nocardia* spiked in clinical sputa[a]

| *Nocardia* species | Culture positivity rate (n/n) | | | MGC[b] (median [IQR], CFU/mL) | | | Growth time (median [IQR], days) | | |
|---|---|---|---|---|---|---|---|---|---|
| | APT | Control | *P* value[c] | APT | Control | *P* value | APT | Control | *P* value[c] |
| *N. wallacei* | 5/7 | 3/7 | 0.480 | $10^3$ | $10^4$ | 0.619 | 3 | 2 | **0.047** |
| (n = 7) | | | | [$10^3$–$10^4$] | [$10^4$–$10^4$] | | [3–3] | [2–2] | |
| *N. mexicana* | 4/5 | 4/5 | | $10^4$ | $10^4$ | | 2 | 2 | |
| (n = 5) | | | | [$10^3$–$10^4$] | [$10^3$–$10^4$] | | [2–2] | [2–2] | |
| *N. otitidiscaviarum* | 3/4 | 1/4 | 0.480 | $10^3$ | $10^3$ | 1 | 2 | 2 | |
| (n = 4) | | | | [$10^3$–$10^4$] | [$10^3$–$10^3$] | | [2–2] | [2–2] | |
| *N. farcinica* | 4/5 | 1/5 | 0.248 | $10^5$ | $10^5$ | 1 | 4 | 5 | 1 |
| (n = 5) | | | | [$10^4$-$10^6$] | [$10^5$–$10^5$] | | [3–4] | [5–5] | |
| *N. cyriacigeorgica* | 3/3 | 2/3 | 1 | $10^3$ | $10^4$ | 0.617 | 3 | 2 | 0.194 |
| (n = 3) | | | | [$10^3$–$10^5$] | [$10^3$–$10^4$] | | [3–4] | [2–2] | |
| *N. abscessus* complex | 1/7 | 1/7 | | $10^3$ | $10^5$ | 1 | 4 | 4 | |
| (n = 7) | | | | [$10^3$–$10^3$] | [$10^5$–$10^5$] | | [4–4] | [4–4] | |
| *N. nova* complex | 2/4 | 1/4 | 1 | $10^6$ | $10^5$ | 1 | 6 | 4 | 1 |
| (n = 4) | | | | [$10^5$–$10^6$] | [$10^5$–$10^5$] | | [6–7] | [4–4] | |
| Total (n = 35) | 22/35 | 13/35 | **0.008** | $10^4$ | $10^4$ | 0.564 | 3 | 3 | 0.081 |
| | | | | [$10^3$–$10^5$] | [$10^3$–$10^5$] | | [2–4] | [2–2] | |

[a]CFU: colony forming unit; APT: amoebae plate test; MGC: minimal growth concentration; IQR: Interquartile range.
[b]MGC is the median of the replicate lowest suspension concentration that gave a positive result in culture.
[c]Significant *P* values are shown in bold.

from the OPF; Table 2; Table S2). Those mixes were tested by APT and control in parallel (example in Fig. 2).

When *Nocardia* serial dilutions were added, the overall rate of positive cultures using APT was almost twice greater (63%, 22/35) compared to control (37%, 13/35, *P* = 0.008; Table 2).

Interestingly, although it did not reach statistical significance, in the control condition, only 1/5 *N. farcinica* grew, while 4/5 grew using the APT. *N. otitidiscaviarum* grew for 1/4 of cases in control vs 3/4 using APT. *N. abscessus* was the most fastidious strain to grow

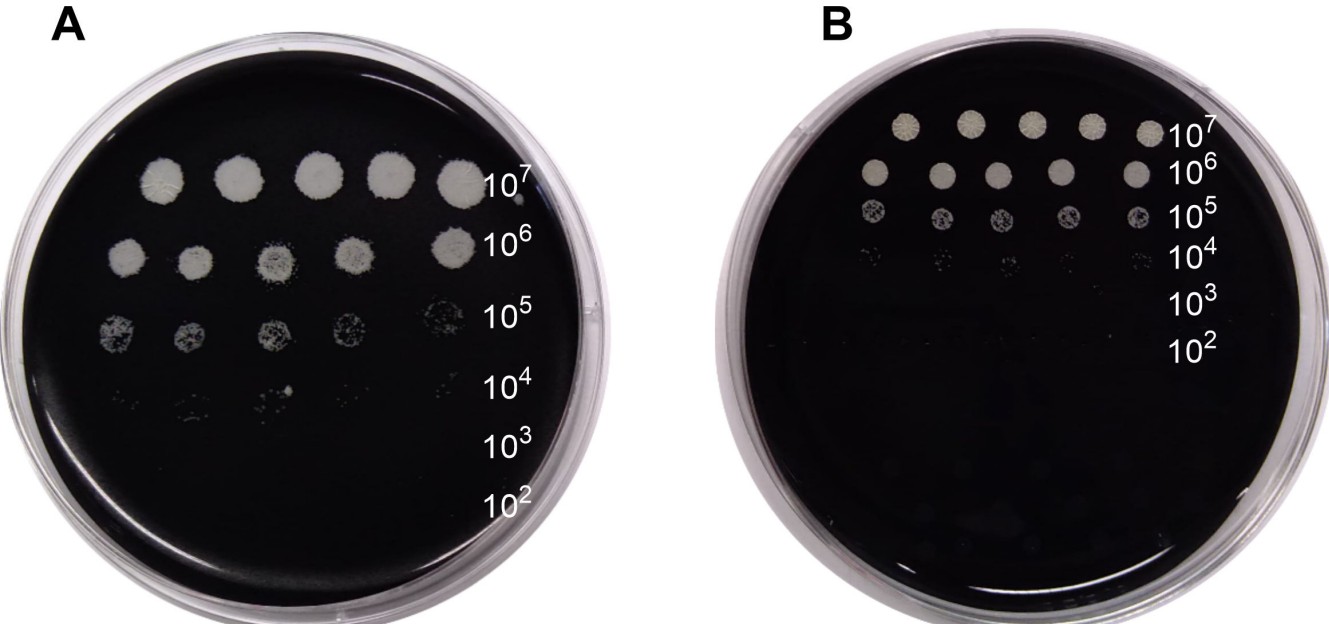

**FIG 1** Example of *N. otitidiscaviarum* (EML 2105) at Day 2 of culture in sterile water. The range of *Nocardia* dilutions is $10^7$–$10^2$ CFU/mL. (A) APT with a positive *Nocardia* growth from $10^7$ to $10^3$ CFU/mL. (B) Control with a positive *Nocardia* growth from $10^7$ to $10^3$ CFU/mL.

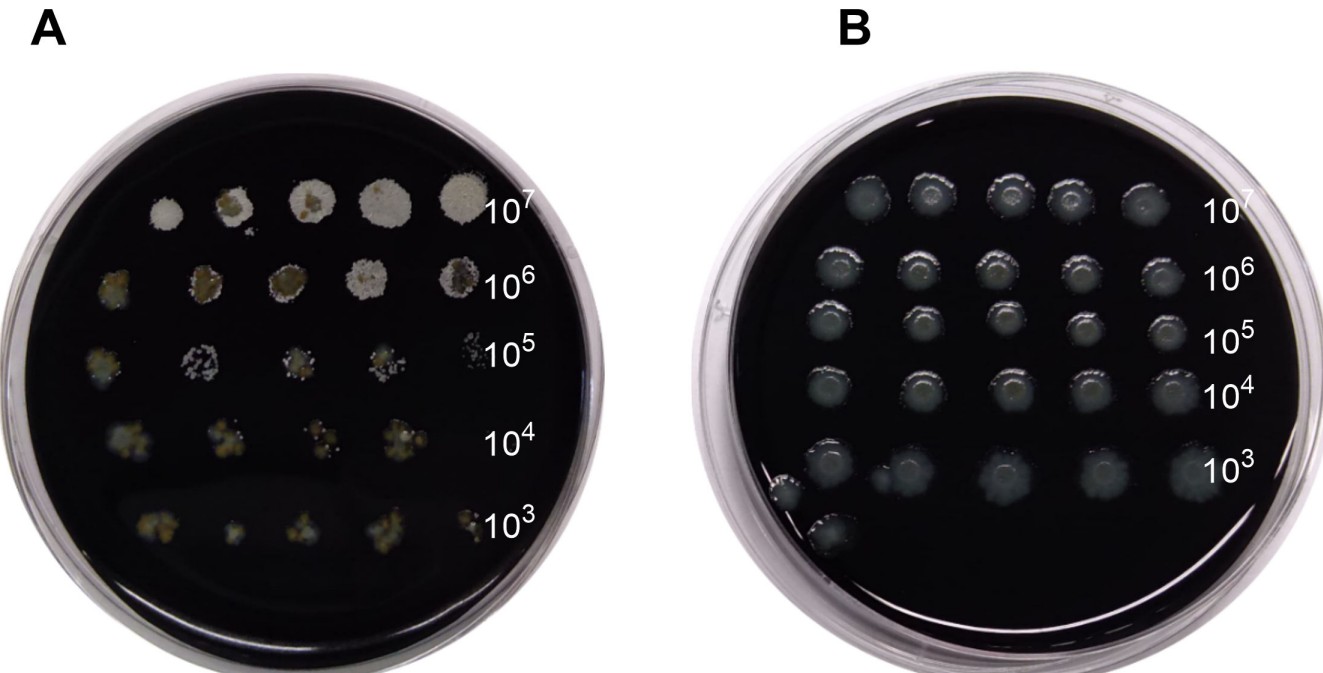

**FIG 2** Example of culture at Day 2 of *N. otitidiscaviarum* (EML 2105) spiked in a sputum. The range of *Nocardia* dilutions is $10^7$–$10^3$ CFU/mL in sputum number 3 (*Pseudomonas aeruginosa* and OPF). (A) APT with positive *Nocardia* growth and medium BF abundance overgrowth in all serial *Nocardia* dilutions. (B) Control with no *Nocardia* growth and high BF growth in all serial *Nocardia* dilutions.

in both the APT (1/7) and control (1/7). *N. nova* was also fastidious to grow in both the APT and control (2/4 and 1/4, respectively). The MGC were not different between the APT and control ($P = 0.564$) and from those found in water ($P = 0.152$ for APT and $P = 0.259$ for control).

Growth using APT was obtained within a median (IQR) of 3 [2–4] days without a significant difference compared to the control ([2–2] days, $P = 0.081$), which was 1 day longer compared to APT in sterile water (2 [2–4]). As in sterile water, disparities in APT growth times were observed between species; *N. mexicana* and *N. otitidiscaviarum* grew within 2 [2–2] days using the APT compared to 4 [4–4] days for *N. abscessus* and 6 [6–7] days for *N. nova*.

## APT enables effective interferent flora reduction

The APT's ability to induce an elimination or a reduction of BF overgrowth in sputa compared to the axenic culture was then assessed. BF growth was significantly reduced using the APT (37%, 13/35) compared to the control (66%, 23/35, $P = 0.004$; Fig. 3A). Similarly, the frequency of *Nocardia* growth without any BF was significantly greater using the APT (34%, 12/35 vs 11%, 4/35, $P = 0.01$). Regardless of *Nocardia* growth, no BF was cultured in 37% (13/35) of the APT compared to 11% (4/35) of the control ($P = 0.007$; Fig. 3B), whereas a high level of BF was found in 23% (8/35) of the APT compared to 63% (22/35) of the control ($P = 0.0005$).

The BF abundance was estimated using an ordinal scale. According to this scale, the median BF abundance of all tested samples was significantly lower using the APT compared to control (1 [0–2] vs 3 [1–3], $P < 0.0001$, Fig. 3C).

## DISCUSSION

The present proof-of-concept study evaluated the APT performance for recovering seven *Nocardia* species clinically relevant and/or at risk of dissemination (*N. abscessus, N. nova, N. cyriacigeorgica*, and *N. farcinica*) (21). All the investigated *Nocardia* species have grown

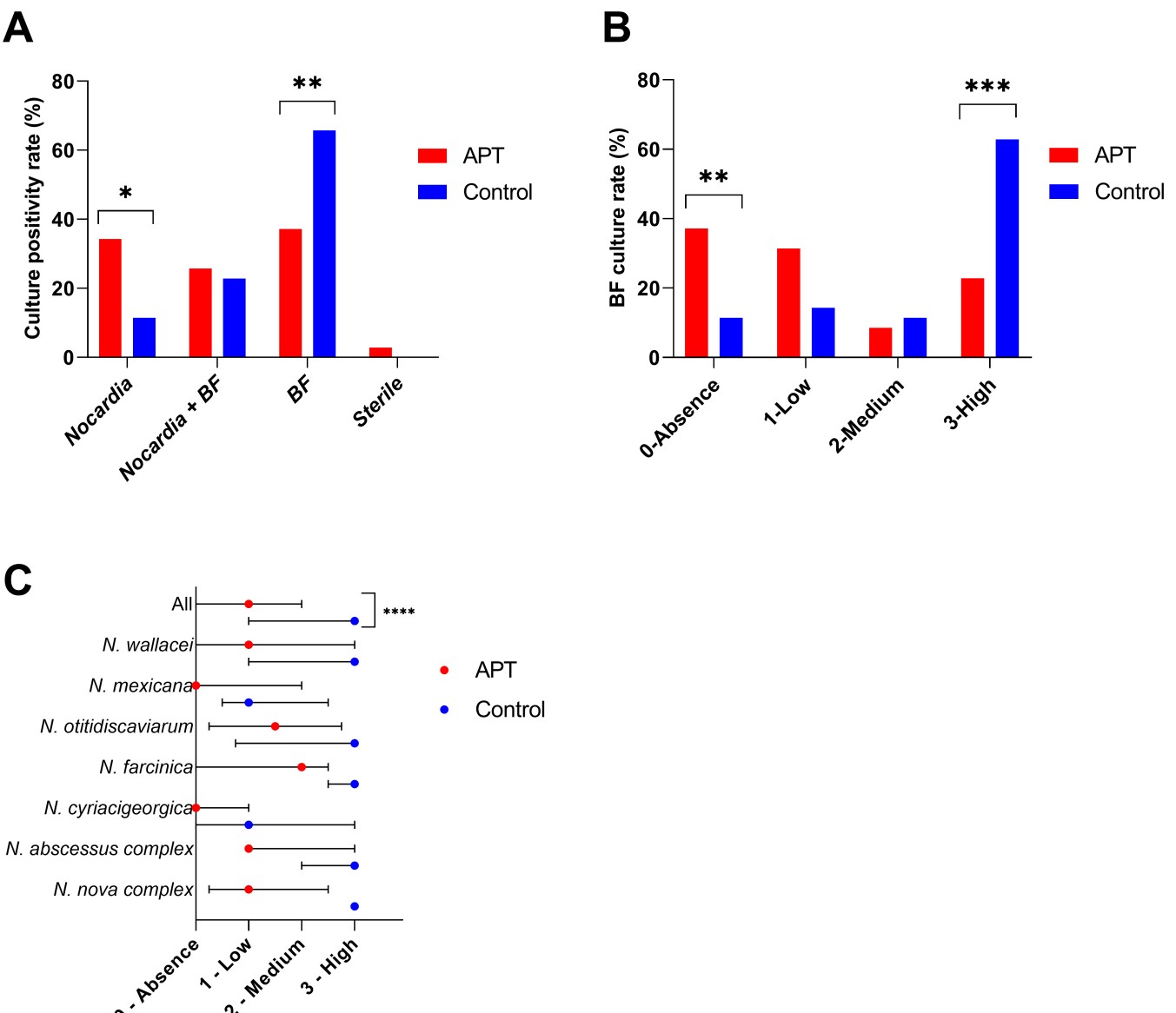

**FIG 3** *Nocardia* and BF culture rates and abundances of BF overgrowth among clinical sputa. (A) *Nocardia* and/or BF culture positivity rate on the amoeba plate test (APT) and axenic culture (control). (B) Culture rate of BF by abundance levels on APT and control, McNemar tests: *$P < 0.05$, **$P < 0.01$, and ***$P < 0.001$. (C) Medians of BF abundance by species on the APT and control. Wilcoxon rank test: ****$P < 0.0001$. Absence: 0 colony; low: one–five colonies; medium: 6–50 colonies; and high: >50 colonies.

in amoebic coculture, but some differences in growth kinetic were found according to the species. Furthermore, *Nocardia* recovery from pulmonary samples containing rich BF was greater, and BF decontamination was more effective using the APT compared to the axenic culture.

In this study, all *Nocardia* species were able to grow across the amoebic monolayer, with overall similar culture positivity rates, MGC, and growth times in water between the APT and axenic culture. This is a strong argument to suggest that *Nocardia* may survive intracellularly within amoebae before cell lysis. Of note, the presence of FLA is unsuitable for the survival of common bacteria because they are usually endocytosed and digested by unicellular protozoans. Yet, some facultative intracellular microorganisms, such as *Legionella*, *Mycobacteria*, and *Stenotrophomonas*, are known or suggested to be hosted by FLA in the environment, allowing them to be protected from hostile environmental conditions (16, 22, 23). For instance, in soils and natural waters, *Legionella* encounter FLA

that represent their natural hosts and reservoir (15, 24). During evolution, *Mycobacteria*, which share a common ancestor with *Nocardia*, developed escape mechanisms to FLA digestion (secretion of toxins, inhibition of phagolysosome fusion) and intra-amoebic replication abilities within several species (25, 26). The evaluation of the intracellular pathways involved in this intra-amoebic development of *Nocardia* was out of the scope of the present study and should be confirmed using *in vitro* amoebic culture models.

The interplay between *Mycobacteria* or *Legionella* and FLA is similar to their interactions with macrophages during a human infection (16, 23). For example, mutants of *M. avium*, which are deficient in some pathogenicity islands, are unable to infect both human macrophages and amoeba of the genus *Acanthamoeba* (16). Similarly, mutants deficient for the Dot/Icm system in *Legionella* infect neither macrophages nor *Acanthamoeba* (27). *Nocardia* have been suggested to be facultative intracellular pathogens. According to Beaman *et al.*, virulent strains of *N. asteroides* adhere to the surface, are internalized, and survive in a non-replicative form inside rabbit lung macrophages or brain cells (11, 12). More recently, *N. seriolae*, a fish pathogen species, has been shown to invade and persist into a murine macrophage lineage (13, 14).

In two independent prospective studies, liquid amoebic coculture and/or APT added to axenic culture have been reported to enhance the *Legionella* recovery rate from microbial overgrown sputum (17, 18); APT is routinely performed as a complementary tool in the *Legionella* French reference center. Accordingly, herein, the *Nocardia* culture positivity rate from sputa was almost two-fold greater when using the APT compared to that when using the control, whereas MCG and growth time were similar. Furthermore, the APT enabled a more frequent elimination and a reduced overall abundance of the BF overgrowth of various bacterial types, regardless of the spiked *Nocardia* species. As for *Legionella*, the APT could then represent a promising second-line tool for enhancing *Nocardia* recovery from overgrown samples, which is a prerequisite to species identification and antimicrobial susceptibility testing.

Some growth differences were observed among species; *N. abscessus* and *N. nova* were the most fastidious and the slowest to grow, respectively. The latter are known as slow growing because their development on solid media is usually obtained within 5–15 days, compared to 2–3 days for other species (9). The 30°C incubation temperature was required to maintain the amoebic culture, whereas the optimal temperature for *Nocardia* and the most common bacteria is 35–37°C. This lower temperature might have delayed the growth of *Nocardia*, especially for slow-growing species. In addition, the BMPA medium containing a second generation cephalosporin (cefamandole) was used because it was the medium tested for the APT initial evaluation (18). *N. abscessus*, *N. nova*, and *N. cyriacigeorgica* were regularly sensitive to third-generation cephalosporins, unlike *N. farcinica* (28). Non-selective (COS) or enriched (BCYE, 7H10) media without cephalosporin may have allowed a better growth of *N. abscessus* and *N. nova*. However, the use of such non-selective medium could likely increase the abundance of BF and interfere with *Nocardia* growth (29).

The present study has some limitations. First, pulmonary Nocardiosis is rare; hence, we could not perform a prospective study on clinical samples and had to artificially overload sputa. Due to the need for fresh samples, we were unable to select those that tested negative for *Nocardia* prior to the study. We did, however, ensure that there was no history of positive *Nocardia* tests in the selected samples. Second, we performed the APT from eight different sputa, generating heterogeneous BF contaminations. Pooling all the sputa could have improved the comparison between the *Nocardia* species growth. In addition, although *Nocardia* suspensions were calibrated after OD measurements, a more precise inoculum quantification, such as enumeration after plating, could have been performed to ensure inoculum reproducibility (30). We did not compare the APT with other decontamination methods commonly used for *Legionella* (heat and acid treatment) or *Mycobacteria* (acid treatment, 4% NaOH solution, or NaOH solution ± N acetylcysteine) (31); these aggressive protocols could reduce *Nocardia* recovery (32). Finally, the 30°C incubation temperature may have slowed the *Nocardia* and BF growth in

the control. The *Nocardia* growth comparison between 30 and 37°C temperatures should be performed using BMPA. Only 30°C was tested; thus, we were not able to evaluate whether the APT would still have increased *Nocardia* isolation yield when compared to the control using bacterial optimal growth temperature.

This study reports *Nocardia's* ability to grow in amoebic coculture, with some growth kinetic differences depending on the species, presumably implying their intra-amoebic multiplication. Furthermore, in APT, *Nocardia* recovery from overgrown sputa was greater, and flora decontamination was more effective. The present findings are strong arguments to implement APT as a complementary technique for *Nocardia* culture in overgrown samples.

## ACKNOWLEDGMENTS

We are grateful to Emmanuelle Bergeron and Betty Ribaut for providing technical assistance with the culture preparation. We are grateful to Shanez Haouari (DRS, *Hospices Civils de Lyon*) for help in the manuscript preparation. We thank Raphaël Fleury for his help with ethical considerations. The graphical abstract (Fig. S1) was created with BioRender.com.

S.J. and F.L. found the main article hypothesis. The project was conceived, planned, and supervised by C.A. and S.J. C.E., G.D., M.I., L.B., C.G.,. J.C., S.J., and C.A. were involved in the design, implementation, and day-to-day management of the study. The *Nocardia* strains were provided by V.R.N. C.E. performed the *Nocardia* culture and APT. V.R.N., and O.D. contributed their expertise on *Nocardia* management. C.E. and C.A. wrote the original manuscript draft, which was reviewed and edited by G.D., M.I., L.B., C.G., O.D., V.R.N., and S.J., and reviewed by all coauthors. Statistical analysis was performed by C.E. and C.A. C.A. produced the graphical abstract. All authors approved the final version of the manuscript.

## AUTHOR AFFILIATIONS

[1]Centre National de Référence des Légionelles, Institut des Agents Infectieux, Hôpital de la Croix-Rousse, Hospices Civils de Lyon, Lyon, France
[2]Centre International de Recherche en Infectiologie (CIRI), Legiopath Team, Inserm, U1111, CNRS, UMR5308, Ecole Normale Supérieure de Lyon, Université Claude Bernard Lyon 1, Lyon, France
[3]ESCMID Study Group for Legionella Infections (ESGLI), Basel, Switzerland
[4]Centre International de Recherche en Infectiologie (CIRI), Stapath Team, Inserm, U1111, CNRS, UMR5308, Ecole Normale Supérieure de Lyon, Université Claude Bernard Lyon 1, Lyon, France
[5]Laboratoire de Biologie Médicale de Référence des Nocardioses, Institut des Agents Infectieux, Hôpital de la Croix-Rousse, Hospices Civils de Lyon, Lyon, France
[6]Centre National de Référence des Staphylocoques, Institut des Agents Infectieux, Hôpital de la Croix-Rousse, Hospices Civils de Lyon, Lyon, France
[7]UMR Ecologie Microbienne, Pathogènes Opportunistes Bactériens et Environnement, CNRS 5557, INRAE 1418, VetAgro Sup, Observatoire Français des Nocardioses, et Université Claude Bernard Lyon 1, Lyon, France

## AUTHOR ORCIDs

Coralie Espinasse ⬤ http://orcid.org/0009-0002-4485-5994
Sophie Jarraud ⬤ http://orcid.org/0000-0001-5750-0215
Camille Allam ⬤ http://orcid.org/0000-0003-4585-4651

## AUTHOR CONTRIBUTIONS

Coralie Espinasse, Formal analysis, Writing – original draft, Writing – review and editing | Ghislaine Descours, Supervision, Writing – review and editing | Marine Ibranosyan,

Validation, Writing – review and editing | Laetitia Beraud, Formal analysis, Methodology, Validation, Writing – review and editing | Christophe Ginevra, Formal analysis, Methodology, Writing – review and editing | Joelle Chastang, Formal analysis, Methodology, Resources, Writing – review and editing | Oana Dumitrescu, Conceptualization, Methodology, Project administration, Resources, Writing – review and editing | Frédéric Laurent, Conceptualization, Investigation, Methodology, Project administration, Resources, Writing – review and editing | Verónica Rodriguez Nava, Conceptualization, Investigation, Methodology, Project administration, Resources, Writing – review and editing | Sophie Jarraud, Conceptualization, Formal analysis, Investigation, Methodology, Project administration, Resources, Supervision, Validation, Writing – original draft, Writing – review and editing | Camille Allam, Conceptualization, Formal analysis, Methodology, Project administration, Supervision, Validation, Writing – original draft, Writing – review and editing

## DATA AVAILABILITY

The data sets analyzed in the present study are available upon reasonable request to the corresponding author.

## ETHICS APPROVAL

The study was approved by an institutional review board (Comité d'éthique des Hospices Civils de Lyon, approval number 23-5280).

## ADDITIONAL FILES

The following material is available online.

### Supplemental Material

**Fig. S1 (Spectrum01416-24-s0001.pdf).** APT assay overview.
**Table S1 (Spectrum01416-24-s0002.pdf).** APT and control MGC and growth times for *Nocardia* in sterile water.
**Table S2 (Spectrum01416-24-s0003.pdf).** APT and control MGC and growth times for *Nocardia*, type, and abundances of BF overgrowth in clinical sputa.

### Open Peer Review

**PEER REVIEW HISTORY (review-history.pdf).** An accounting of the reviewer comments and feedback.

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
