## [Reviewer comments · Microbiology Spectrum]

Microbiology Spectrum

Amoeba Plate Test (APT) with *Acanthamoeba castellanii* as an innovative tool for *Nocardia* recovery from sputum samples: a proof-of concept study

Coralie Espinasse, Ghislaine Descours, Marine Ibranosyan, Laëtitia Beraud, Christophe Ginevra, Joelle Chastang, Oana Dumitrescu, Frédéric Laurent, Veronica Rodriguez-Nava, Sophie Jarraud, and Camille Allam

Corresponding Author(s): Sophie Jarraud, Hospices Civils de Lyon

Review Timeline:

Submission Date:	June 18, 2024
Editorial Decision:	July 23, 2024
Revision Received:	September 25, 2024
Accepted:	October 15, 2024

Editor: Florence Doucet-Populaire

Reviewer(s): Disclosure of reviewer identity is with reference to reviewer comments included in decision letter(s). The following individuals involved in review of your submission have agreed to reveal their identity: Lucio Vera-Cabrera (Reviewer #2)

Transaction Report:

DOI: <https://doi.org/10.1128/spectrum.01416-24>

Re: Spectrum01416-24 (Amoeba Plate Test (APT) as an innovative tool for *Nocardia* recovery: a proof-of concept study)

Dear Dr. Sophie Jarraud:

Thank you for the privilege of reviewing your work. Below you will find my comments, instructions from the Spectrum editorial office, and the reviewer comments.

Revision Guidelines

Sincerely,
Florence Doucet-Populaire
Editor
Microbiology Spectrum

Reviewer #1 (Comments for the Author):

Comment on « Amoeba Plate Test (ATP) as an innovative tool for *Nocardia* recovery: a proof of concept study"

In the study, the authors used Amoeba Plate Test in order to improve the sensitivity of *Nocardia* detection. This test is already used for the detection of *Legionella* in clinical samples.

The topic is of interest as the detection of *Nocardia* in clinical samples is challenging.

These results are preliminary because they assessed their ability to detect a standardized inoculum of *Nocardia* mixed in sterile water or in *ex vivo* human sputum.

However, if confirmed in clinical studies, this approach could significantly improve the management of patients.

In this perspective, this paper should be considered as a significant step forward into the field.

Some major comments should be taken into account

Major comments

The authors claim (even in the "Importance" section) that their data provide "strong argument that *Nocardia* may survive inside amoeba". However, this was not the topic of this paper and the authors did not assess this point. I would be more balanced and recommend not including this in the "Importance" section. This question is relevant in the discussion section (where it is already mentioned).

From the ethical point of view, it is surprising that patients were not informed of the use of their sputa for research purpose. This use is clearly outside the scope of a clinical sample. Changing the final use of a human sample should require, at the very least, information to the patients.

The concept of MGC is not clear and not detailed enough. Maybe, it should be called: minimal bacterial concentration allowing detection. Or at least better defined and explained

Minor comments

Some sentences should be rephrased and the paper would benefit from corrections by a native English writer.

Introduction: the authors state that pulmonary nocardiosis is under reported. Can they provide robust data to support this statement?

Introduction, line 87, I suggest adding a connection between these two sentences: "On the other hand, it is well established..."

Reviewer #2 (Comments for the Author):

The present work deals with an assay to isolate *Nocardia* species from sputum samples using an amoeba co-culture system in order to decrease bacterial contamination. The manuscript is well written, in an appropriated English style. My comments are the following:

1. Tilt, Put the word "*Nocardia*" in italics
2. You may add "recovery from sputum samples"
3. You need to add in the title the amoeba species used (*Acanthamoeba castellanii*)
4. Line 48, Please change "coculture" for "co-culture"
5. Line 52, change to: "...*Nocardia* isolation from heavily contaminated samples"
6. Line 115, *Nocardia* usually grows as entangled filaments. How did you prepare the suspensions? Did you disaggregate the clumps? Did you first prepare a culture in broth? The quantification by measuring the O.D. is not very accurate. You need to plate the suspensions and count them.
7. Paragraph on line 143, Did you test the antimicrobials used for activity on the *Nocardia* species utilized?
8. Line 280, you need to run the test at 37°C in order to rule out the growth rate.
9. References section needs to be re-written, there are many errors. You need to write all the microorganisms' names in italics. To put all the titles in the correct format (first word in capital and the rest in low letters. Reference 16 is incomplete.

Reviewer 1, Comment 1. The authors claim (even in the "Importance" section) that their data provide "strong argument that *Nocardia* may survive inside amoeba". However, this was not the topic of this paper and the authors did not assess this point. I would be more balanced and recommend not including this in the "Importance" section. This questions is relevant in the discussion section (where it is already mentioned).

Authors: As requested, we have withdrawn this sentence from the importance section.

Reviewer 1, Comment 2. From the ethical point of view, it is surprizing that patients were not informed of the use of their sputa for research purpose. This use is clearly outside the scope of a clinical sample. Changing the final use of a human sample should require, at the very least, information to the patients.

Authors: The samples involved in this project were collected from de-identified (irreversibility rendered anonymous) residual routine samples from patients hospitalized at the Hospices Civils de Lyon (HCL). Regarding the use of residual data/biological samples for research purposes, HCL has implemented a global information procedure: At the time of admission, each patient is informed of the potential use of his or her biological samples for research purposes, and of his or her right to oppose such use in the context of health care, by means of an information leaflet.

As the samples used were rendered irreversibly anonymous, a waiver of the obligation to individually inform the patient was requested from the Institutional Review Board.

The study design and this specific request were submitted to and approved by the Institutional Review Board under number 23-5280.

Reviewer 1, Comment 3. The concept of MGC is not clear and not detailed enough. Maybe, it should be called: minimal bacterial concentration allowing detection. Or at least better defined and explained

Authors: You are right, MGC was only detailed in table 1 and table 2 legends. A clarified definition was added in material and methods section lines 165-167: “Day of culture positivity and minimal growth concentration (MGC) were assessed for each replicate on APT and control media. MGC was defined as the lowest suspension concentration that gave a positive result in culture.”

Reviewer 1, Comment 4. Some sentences should be rephrased and the paper would benefit from corrections by a native English writer.

Authors: Thank you for this remark. To enhance the quality of the English, this revised version of the manuscript has been thoroughly proofread by a dedicated department in our institution, specialized in English scientific proofreading (DRS, Direction de la Recherche Scientifique, Hospices Civils de Lyon).

Reviewer 1, Comment 5. the authors state that pulmonary nocardiosis is under reported. Can they provide robust data to support this statement?

Authors: Nocardiosis is regularly described in the literature as an underreported disease. According to Sun *et al*, (Front Cell Infect Microbiol 2023, reference 1 in the manuscript) and Saubolle *et al*, (J Clin Microbiol. 2003, reference 4 in the manuscript) due to the lack of *Nocardia* screening methods, *Nocardia* is often missed in diagnosis, leading to worsening the

condition. Furthermore, the varied clinical presentations can make clinical recognition challenging Traxler *et al*, Clin Microbiol Rev 2022, and reference 2 in the manuscript. Metagenomic next generation sequencing (mNGS) has shown value in the diagnosis of Nocardiosis due to its non-targeted approach. Liang *et al* (BMC Infectious Diseases 2023, reference 3 in the manuscript) analysed 3756 respiratory and blood samples by mNGS, of which 19 were positive for *Nocardia*. Among them, 12 patients showed pneumonia compatible with pulmonary nocardiosis. In none of the 12 patients was the diagnosis suspected. Culture was retrospectively positive in only 2 specimens. Nocardiosis may be underdiagnosed or misdiagnosed, especially in an immunocompetent or geriatric population as Liang *et al*. For clarification, we added the following sentence lines 71-73: “The lack of *Nocardia* screening methods (1,4) as well as the varied clinical presentation (2) are the main causes of Nocardiosis underdiagnosis” and three additional references 1, 2, 4 line 73.

Reviewer 1, Comment 6. Introduction, line 87, I suggest adding a connection between these two sentences: "On the other hand, it is well established..."

Authors: Revised accordingly (line 87).

Reviewer 2, Comment 1. Title, Put the word "*Nocardia*" in italics. You may add "recovery from sputum samples". You need to add in the title the amoeba species used (*Acanthamoeba castellanii*)

Authors: Revised accordingly: “Amoeba Plate Test (APT) with *Acanthamoeba castellanii* as an innovative tool for *Nocardia* recovery from sputum samples: a proof-of concept study”.

Amoeba species was added as well in abstract line 36.

Reviewer 2, Comment 2. Line 48, Please change "coculture" for "co-culture"

Authors: We apologize for this error that has been revised (line 48).

Reviewer 2, Comment 3. Line 52, change to: "*Nocardia* isolation from heavily contaminated samples".

Authors: Revised accordingly (line 52).

Reviewer 2, Comment 4. Line 115, *Nocardia* usually grows as entangled filaments. How did you prepare the suspensions? Did you disaggregate the clumps?

Authors: *Nocardia* colonies were sampled with a swab from fresh culture on solid media then suspended in sterile water. The suspension was homogenized by vortexing for 30s and pipetting up and down with a micropipette. A particular attention was paid to obtain a homogeneous, precipitate-free suspension prior to OD measurement. Homogenization procedure was repeated between OD adjustments (e.g. when the suspension was re-loaded with *Nocardia* from solid media). Vortex homogenization and pipetting up and down were performed between each dilution points. Precisions on homogenization were added in the materials and methods section lines 117-120 to enhance the clarity of the document: "Suspensions were homogenized prior to OD measurement and between the dilution points by vortexing for 30s and pipetting. Since *Nocardia* usually grow as entangled filaments, a particular attention was paid to obtain a homogeneous, precipitate-free suspension prior to OD measurements".

Reviewer 2, Comment 5. Did you first prepare a culture in broth? The quantification by measuring the O.D. is not very accurate. You need to plate the suspensions and count them.

Authors: *Nocardia* were grown on solid media only (5% blood sheep agar). We used a quantification by OD measurement as described in the APT protocol Descours *et al*, 2018

(17). Our objective was to evaluate the performance of APT in comparison with axenic culture. Therefore, both techniques were evaluated in parallel using the same *Nocardia* suspensions for APT and control to ensure comparability between them. However, inoculum variability may exist between replicates. An enumeration by culture of the suspensions, as performed by Conville *et al*, 2012 (30) could have been performed to ensure reproducibility of the inoculum between replicates. Consequently, we added the limits on inoculum quantification to the discussion, lines 302-304: “In addition, although *Nocardia* suspensions were calibrated after OD measurement, a more precise inoculum quantification such as enumeration after plating could have been performed to ensure inoculum reproducibility (30).”

Reviewer 2, Comment 6. Paragraph on line 143, Did you test the antimicrobials used for activity on the *Nocardia* species utilized?

Authors: Antimicrobials were not tested on *Nocardia*. Polymyxin B and anisomycin are inactive against *Nocardia*. Gram-positive bacteria are naturally resistant to polymyxin B, and anisomycin is included in BMPA media for its antifungal properties. However, cefamandole may have inhibited *Nocardia* growth. This point is discussed in lines 288-294: “In addition, the BMPA medium containing a second-generation cephalosporin (cefamandole) was used since it was the medium tested for APT initial evaluation (18). *N. abscessus*, *N. nova*, and *N. cyriacigeorgica* have been found to be regularly sensitive to third generation cephalosporins, unlike *N. farcinica* (28). Non-selective (COS) or enriched (BCYE, 7H10) media without cephalosporin may have allowed a better growth of *N. abscessus* and *N. nova*. However, the use of such non-selective medium could likely increase the abundance of BF and interfere with *Nocardia* growth (29).”

Reviewer 2, Comment 7. Line 280, you need to run the test at 37C in order to rule out the growth rate.

Authors: We were unable to test the APT at a temperature of 37°C because the amoebic layer would no longer be viable at this temperature. As explained in the discussion, “The 30°C incubation temperature was required to maintain the amoebic culture whereas the optimal temperature for *Nocardia* and most common bacteria is 35-37°C. This lower temperature might have delayed the growth of *Nocardia* especially for slow growing species”; lines 285-288. In order to compare *Nocardia* counts between control and APT, the 30°C temperature was also used for the control BMPA plates. We discussed this limitation lines 307-308: “the 30°C incubation temperature may have slowed the *Nocardia* and BF growth in the control. Since only this temperature was tested, we could not evaluate whether the APT would still have increased *Nocardia* isolation yield when compared to control using bacterial optimal growth temperature”. We added the following sentence: “*Nocardia* growth comparison between 30°C and 37°C temperatures should be performed using BMPA” lines 309-310.

Reviewer 2, Comment 8. References section needs to be re-written, there are many errors. You need to write all the microorganisms’ names in italics. To put all the titles in the correct format (first word in capital and the rest in low letters. Reference 16 is incomplete.

Authors: Revised accordingly.

Re: Spectrum01416-24R1 (Amoeba Plate Test (APT) with *Acanthamoeba castellanii* as an innovative tool for *Nocardia* recovery from sputum samples: a proof-of concept study)

Dear Dr. Sophie Jarraud:

Your manuscript has been accepted, and I am forwarding it to the ASM production staff for publication. Your paper will first be checked to make sure all elements meet the technical requirements. ASM staff will contact you if anything needs to be revised before copyediting and production can begin. Otherwise, you will be notified when your proofs are ready to be viewed.

Sincerely,
Florence Doucet-Populaire
Editor
Microbiology Spectrum

Reviewer #1 (Comments for the Author):

The authors adequately took into account all my comments

Reviewer #2 (Comments for the Author):

I still consider that the suspensions prepared by vortexing the nocardia cultures are not sufficiently homogenous and that is very important to standardize in your assay. The presence of entangled filaments or cumuli of bacterial cells will alter the intake by the amoeba, and therefore modify your results. Unicellular suspensions must be prepared by grinding the Nocardia culture using a Potter-Evelham device and separate the unicellular suspension from the aggregates by centrifugation at 100xg, and then count by agar plating, in order to ensure a better control of the assay. Please check the technique in : Welsh-Lozano O, Rodríguez M A, Salinas-Carmona M C, Vera-Cabrera L. Experimental mycetoma by *N. brasiliensis* in rats. J Mycol Med. 1998;8:183-187. I still consider that the suspensions prepared by vortexing the nocardia cultures are not sufficiently homogenous and that is very important to standardize the assay. Nocardia cultures are hard to disaggregate, and they need to validate their results by

preparing a unicellular suspension of the Nocardia species tested and run an assay comparing with the suspensions prepared by vortexing the cultures. Much of the results depend on the ingestion of the Nocardia by the amoebas, that may be different if presented in cumuli.